# Forms of Bacterial Survival in Model Biofilms

**Timofei A. Pankratov [1,*], Yuri A. Nikolaev [1,2], Yulia K. Yushina [2], Ekaterina N. Tikhonova [1] and Galina I. El-Registan [1]**

1 Federal Research Center "Fundamentals of Biotechnology" of RAS, 117312 Moscow, Russia
2 V.M. Gorbatov Federal Research Center for Food Systems of RAS, 109316 Moscow, Russia
* Correspondence: tpankratov@gmail.com; Tel.: +7-(499)-135-01-80

**Abstract:** Bacterial survival upon sharp fluctuations of environmental parameters and exposure to lethal doses of stressors (antibiotics, disinfectants, heat shock, and others) is ensured by the use of different strategies of resistance, an important place among which is occupied by the forms with reduced or stopped metabolism, antibiotic tolerant (AT) persister (P) cells, anabiotic dormant forms (DFs), and viable but non-culturable (VBNC) cells. Elucidating the role of these forms of bacterial resistance to an impact of chemical and biological toxicants and physical stressors is of great fundamental and practical interest. The aim of this research was to study the dynamics of the resistance forms in bacteria developing in biofilms and, for comparison in liquid media, upon exposure to lethal doses of antibiotics and heat shock (80 °C, 15 min). In the trials, the experimental model of the development of monospecies and binary forms of bacterial biofilms including contaminants of meat products (eight strains of genera *Pseudomonas*, *Escherichia*, *Salmonella*, *Staphylococcus*, and *Kocuria*) on the fiberglass filters was used. It was established that survival of populations in the presence of lethal doses of antibiotics and upon heating was ensured by persister cells forming in bacterial populations and, at the late stages of the development of biofilm or planktonic cultures (28 days of incubation), by anabiotic DFs. With that, the number of thermoresistant (TR) DFs ($10^3$–$10^4$ CFU/mL) in dying biofilms (28 days) developing in the standard conditions (composition and volume of a medium, pH, growth temperature) weakly depended on the bacterial taxonomic status. This study demonstrates the heterogeneity of DF populations of biofilm bacterial cultures in terms of the depth of dormancy, as a result of which the number of thermoresistant DFs after heating can exceed their total number before heating (due to the effect of DF revival, resuscitation). When studying the dynamics of TR cells and P cells in bacterial biofilm and planktonic cultures, it was found that their number (CFU/mL) in populations decreased up to the absence of TR cells and P cells on the 21st day of growth and was restored on the 28th day of growth. The revealed phenomenon can be explained only by cardinal changes in the ultrastructural organization of cells, namely, cytoplasm vitrification due to a sharp decrease in an amount of free water in a cell, which, according to the results obtained, occurs in the period between the 21st and 28th days of incubation. A high degree of correlation between the number of AT cells and TR cells (0.5–0.92) confirms the hypothesis that regards P cells as precursors of DFs.

**Keywords:** bacteria; food processing; mono- and multi-species biofilms; survival forms; persisters; dormant forms

## 1. Introduction

Survival of microbial populations in the environment depends on the adequacy of the microbial cell metabolism, which is controlled by regulatory systems, environmental conditions, determining homeostasis of a cell, and the population in general. The main form of microbial existence in the natural systems is their growth in biofilms (BFs) [1–3]. Biofilms are mono-species or multi-species communities embedded into the polymeric material-matrix produced by microorganisms themselves. Such growth in biofilms facilitates microbial survival in conditions that are constantly changing but suitable for growth.

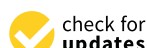



Living organisms (humans, animals, plants) as well as object of human business activities are also colonized by microorganisms (saprophytes, commensals, and pathogens), developing as biofilms.

The microbial biofilm growth in food industry enterprises poses a serious threat not only for the food industry but also for medicine, as food enterprises can be reservoirs for the development and spread of pathogenic bacteria [4–9].

Methods for sanitary treatment applied in food enterprises do not ensure a sufficient level of sterility and, thus, microbiological safety [9–12]. This situation is explained by the high resistance of BFs to damaging impacts, which is associated with: (1) the protective function of the biofilm matrix [13,14]; (2) increased resistance of the biofilm phenotype of cells in the BF composition [15,16]; (3) an increase in the resistance of cells in multi-species BFs compared to mono-species [6,7,17–20]; and (4) age-related heterogeneity of the biofilm population, where old cells constitutively have higher adaptive resistance to sterilization means and other damaging agents than younger cells [12,21,22]. However, the question of physiological heterogeneity of cells in BFs, that is, the presence, proportion, and balance of different cell types (active cells, persisters, and dormant forms), is not investigated in detail. Several studies note the high content of persister cells in BFs [23,24].

As BFs (including food BFs) develop, conditions preventing the further growth of the population (exhaustion of nutrient sources, critical cell density in the population) are created in them. In this case, the BF growth strategy is replaced by the strategy of its survival due to formation of dormant forms (DFs) of cells highly resistant to stress. These cells are resistant to the action of disinfectants and antibiotics, as well as to acidification or alkalization of the environment and heat treatment [25–27], that is, to treatments used in disinfection.

Discovery of the phenomenon of bacterial cell persistence and its intensive study today have extended the notion of microbial adaptation and their survival upon lethal exposure [28–30]. It has been shown that bacterial populations growing planktonically or as biofilms form a small (0.001%–0.1%) subpopulation of persister (P) cells during their development, which increases by several orders of magnitude (1%–5%) in the stationary growth phase upon exhaustion of nutrient sources or the critical cell density in the population. As a result of cytodifferentiation, these cells form a survival phenotype in growth-limiting conditions. This leads to formation of non-dividing or very slowly dividing cells resistant to lethal doses of antibiotics as well as disinfectants, heavy metal ions, acidification, or alkalization of the environment and other stressors. Such cells revert to division upon termination of lethal exposure in a fresh medium and reproduce the parent population, where persister cells will also form [28,31–33].

Therefore, the phenomenon of persistence fits well into the "dormancy continuum" [34] as a form of the temporally metabolically inactive condition along with dormant (anabiotic) forms (DF) and viable but nonculturable (VBNC) cells that ensure survival of the population in conditions unfavorable for growth [35–37].

Based on the foregoing, the aim of the present research was to study forms of high resistance in model biofilms (mono-species and binary) of opportunistic bacteria—meat product contaminants—to lethal doses of biocides (antibiotics) and heat shock. The tasks of the research included: (1) to study the development of persister cells in the dynamics of formation and aging of biofilms of opportunistic bacteria, including those that were isolated from meat products in the meat processing plant Cherkizovsky (Moscow, RF), formed on fiberglass filters; (2) to determine a level of heat resistance of developing and aging biofilms and a possible correlation between them; (3) to study the ultrastructural organization of cells surviving in biofilms incubated for a long time.

## 2. Materials and Methods

### 2.1. Bacterial Cultures

In this work, cultures of gram-positive and gram-negative bacteria were used: *Salmonella typhimurium* TA 1535, *Escherichia coli* K12, *Pseudomonas aeruginosa* S481, *P. extremorientalis*

S85, *P. antarctica* S222, *Staphylococcus aureus* ATCC6538, and *Kocuria rhizophila* S155. Cultures of *P. extremorientalis* S85, *P. antarctica* S222, and *K. rhizophila* S155 were isolated from meat products and identified by the MALDI-TOF method in the V.M. Gorbatov Federal Research Center for Food Systems of the Russian Academy of Sciences by standard methods [38]. The rest of the strains were obtained from the UNIQEM collection (Research Center for Biotechnology of the Russian Academy of Sciences). Bacteria were cultivated in LB (Luria-Bertani) liquid and solid media (LB Broth, Miller).

### 2.2. Production of Planktonic Populations

To prepare the inoculum, isolated bacterial colonies were picked up from LB agarized medium. Biomass of two to three individual colonies was transferred into 20 mL of the sterile LB liquid medium in 50 mL flasks and incubated in a rotary shaker (150 rpm) at 30 °C. To obtain test cultures, an inoculant, which was a culture at the beginning of the stationary phase (overnight culture), was transferred into 20 mL of the LB medium in an amount ensuring the initial number of the test culture of ~$8.0 \times 10^7$ cells/mL. Incubation was carried out upon agitation in the rotary shaker (150 rpm) at 30 °C.

To determine the number of viable cells (CFU/mL), samples of developing cultures (100 µL) were taken every seven days during one month in the aseptic conditions, and serial dilutions in the sterile isotonic sodium chloride solution (0.9%) were prepared. Plating from the corresponding dilutions was carried out by the modified micro-method [39]. To this end, 5 µL of the suspension was taken from the obtained serial dilutions and inoculated (a drop) into the agar medium in Petri dishes (5–10 drops per variant, number of replicates: 5–7). In all cases, the total number of cells (TNC) was defined as the number of viable cells capable of forming isolated colonies (CFU).

### 2.3. Preparation of Experimental Biofilms

The method of biofilm formation on the fiberglass filters was taken as a basis [40]. Filters (225 mm$^2$) were cut from commercial fiberglass paper (Whatman GF/F) and sterilized by autoclaving (20 min, 121 °C). Then, filters were placed on the surface of the LB agar medium in Petri dishes. Overnight planktonic cultures were diluted with the sterile LB liquid medium up to an optical density of 0.5 ($\lambda = 540$ nm), and a 20 µL aliquot was placed on the fiberglass filters and uniformly distributed over the filter surface. After plating, Petri dishes were turned upside down and incubated at 28 °C for four weeks, controlling the number of cells (CFU/mL) on days 2, 7, 14, 21, and 28. To obtain binary biofilms, aliquots of mono-species planktonic cultures were pooled (500 µL each), and the obtained mixture was inoculated on filters as described above. Culture pairs for binary biofilms were selected experimentally, taking into account the colonial morphology and growth rate compatibility: *E. coli* + *K. rhizophila*; *S. typhimurium* + *S. aureus*.

### 2.4. Determination of the Number of Viable Cells in Biofilms

To determine the number of cells in biofilms, fiberglass filters with biofilms grown on them were initially homogenized (homogenizer Ultra Turax, IKA, Staufen, Germany) in 10 mL of the sterile physiological solution in DT-20 tubes with the rotor-stator element in the following mode: 10 s at 4000 rpm, 10 s at 6000 rpm, and the rest of the time up to 2 min at 8000 rpm. Then, 100 µL of homogenate were transferred into 900 µL of the sterile physiological solution, and serial dilutions were prepared. The number (CFU/mL) was determined by the micro-method as described above and calculated per filter (225 mm$^2$) (number of replicates: 5–7). Colony counts in mixed (binary) seeding were performed based on differences in the shape, color, and consistency of the colonies.

### 2.5. Determination of the Proportion of Thermoresistant Cells in Planktonic Cultures and Biofilms

To determine the content of thermoresistant cells in developing planktonic cultures, 500 µL of the culture was taken and heated in 2 mL Eppendorf tubes at 80 °C for 15 min. Then, serial dilutions were prepared, and the number of CFU/mL was determined as

described above. A proportion (%) of thermoresistant cells was determined by a ratio of the number of cells that survived after heating to the number of cells before heating (100%).

To determine the content of thermoresistant cells in a biofilm population grown on a fiberglass filter, the method for assessment of the cell number in the biofilm homogenate (500 μL) before and after its heating was used as described above.

### 2.6. Determination of the Number of Antibiotic Tolerant Cells in Biofilms

To determine the number of antibiotic tolerant cells (ATCs), to which persister cells are assigned, the method for their selection by antibiotics was used [41]. In the following, the term persisters (P) is used as a synonym for AT cells. Antibiotic ciprofloxacin (CF) (10 μg/mL) was used for gram-negative bacteria and chloramphenicol (CP) (25 μg/mL) for gram-positive bacteria, which corresponded to 10 MIC. To determine the number of P cells, biofilms of different age (2–28 days of growth) were homogenized as described above. The homogenate (100 μL) and the sterile LB liquid medium (900 μL), to which a solution of one of the antibiotics was added beforehand, were transferred into Eppendorf tubes. The control samples did not contain antibiotics. Then, the Eppendorf tubes with the suspension (1000 μL) were incubated at 34 °C for seven hours upon agitation at 750 rpm in a thermoshaker (Biosan TS-100, Riga, Latvia). At regular intervals, an aliquot (100 μL) was taken from suspensions, and serial dilutions in the physiological solution were prepared. Plating and counting of the number of CFU/mL were carried out as described above. In the text below, the number of CFU on a filter area of 225 mm$^2$ is given.

### 2.7. Scanning Electron Microscopy

Bacterial biofilms grown on fiberglass filters were fixed with glutaraldehyde and dehydrated in a series of ethanol solutions by the following scheme: glutaraldehyde, 2.5%—90 min with the following washing with the phosphate buffer; ethanol 30%—2 min; 50%—7 min; 70%—5 min; 95%—5 min. After that, a fragment of a fiberglass filter with a biofilm was fixed on the substrate (with double sided tape) attached to a copper cylinder. Gold sputtering was carried out using a Jeol Fine Coat Ion Sputter JFC-1100 (Jeol, Tokyo, Japan). Then, cylinders were loaded into the working chamber of a JEOL JSM-IT200 Scanning Electron Microscope (Jeol, Tokyo, Japan), and monitoring of the sample surfaces was carried out using the bundled software.

### 2.8. Transmission Electron Microscopy

To study the ultrastructural organization of cells, a JEOL JEM-1400 Transmission Electron Microscope (Tokyo, Japan) (accelerating voltage: 120 kV, magnification: 10,000× to 35,000×) was used. Precipitated cells of planktonic or biofilm cultures were fixed in the 1.5% glutaraldehyde solution in 0.05 M cacodylate buffer and further fixed in the 1% OsO4 solution in 0.05 M cacodylate buffer (pH 7.2) at 20 °C for three hours. After dehydration, material was embedded in epoxy resin Epon 812. Ultrathin sections were contrasted in the 3% uranyl acetate solution in 70% ethanol for 30 min and further stained with Reynolds' lead citrate at 20 °C for 4–5 min, and viewed under a microscope.

### 3. Results

Planning of this experiment envisaged elucidation of the causes of high stress resistance of bacterial biofilm cultures compared to planktonic ones, which is widely studied in medical and biotechnological aspects. The present study emphasizes the role of age-related heterogeneity of biofilms, namely the coexistence of not only dividing vegetative cells, but also stationary and dormant ones, as one of the causes of stress resistance in BFs. Earlier, coexistence of vegetative (dividing and stationary) cells and dormant forms (DF) with the specific ultrastructural organization was shown in BFs isolated in meat processing plants [42].

The present study highlights resistance of bacteria in biofilms to lethal exposure to antibiotics and heat treatment (80 °C, 15 min) in the process of the BF development and



aging due to formation and survival of stress-resistant cells. For comparison, formation of stress-resistant (thermoresistant) cells was studied in planktonic bacterial cultures incubated at constant agitation for a long time (28 days).

### 3.1. Dynamics of Surviving and Thermoresistant Cells in Developing Planktonic Bacterial Cultures

It is known that the developmental cycle of planktonic bacterial cultures ends with formation of a small subpopulation of dormant forms (DFs) represented by endospores in spore-forming bacteria [43] and cyst-like cells (CLC) in non-spore-forming bacteria [25,27,44]. One of the determining characteristics of dormant forms is their high stress resistance, including thermal resistance [45,46]. DFs forming at the final stage of the population development are dedicated to its long-term survival (up to millions of years). The vegetative forms that survive lethal exposures are persisters (P), whose determining feature is resistance to lethal doses of antibiotics [28,29,41], as well as other stressors, including high temperatures.

The dynamics of the development (28 days) of planktonic cultures of both gram-negative and gram-positive bacteria was characterized by the uniformity of their transition to the stationary phase after 24 h of growth with the subsequent dying of the population. Populations of *P. extremorientalis*, *P. antarctica*, and *P. gessardii* lost their culturability most rapidly (Figure 1a). In the *P. aeruginosa* culture, the number of colony forming cells stabilized ($10^5$–$10^6$ CFU/mL) after 14 days of rapid loss of cultivability of the population and remained unchanged up to the end of the experiment (Figure 1a).

In the cultures of the *Enterobacteriaceae* members (Figure 1b) as well as in the cultures of gram-positive bacteria (Figure 1c), rapid dying of populations was observed up to the seventh day of growth; then, the death rate was sharply reduced, and after 28 days the population size was about $10^5$ CFU/mL.

The number of thermoresistant cells (TRCs) in the stationary cultures of the studied bacteria (*P. aeruginosa* (Figure 1d), *E. coli*, and *S. typhimurium* (Figure 1e) and *K. rhizophila* (Figure 1f) was about 1% of the total number of their populations, which corresponds to the number of type I persisters developing in the stationary growth phase of the planktonic bacterial cultures [32,41].

The following dynamics of the number of TRCs studied for the first time was characterized by an unexpected effect: a reduction in the number of TRCs with the minimum on the 21st day of incubation and a significant increase in their number in bacterial cultures on the 28th day. This effect was most pronounced for *P. aeruginosa* (Figure 1d), as well as for *S. aureus* and *K. rhizophila* (Figure 1f) compared to the dynamics of the TRCs of the *Enterobacteriaceae* members (Figure 1e). It is worth noting that the proportion (%) of thermoresistant dormant cells relative to the total number of cells (TNC) on this day of incubation varied depending on species and time of incubation. The most interesting effect was observed in case of *P. aeruginosa*: the proportion of TRCs in the composition of the surviving population on the 14th and 28th days of incubation exceeded TNC revealed by the plating method (Table 1).

The values of the proportion (%) of TRCs relative to the total number of cells in the surviving population are given in Figure 2 for the bacterial cultures that are most significant in terms of sanitary welfare of food enterprises. The proportion of TRCs is similar to the dynamics of their number (Figure 1d–f): the highest proportion of TRCs was found in the 7-day cultures, the minimum proportion in the 21-day cultures, with an increase in the TRC proportion in 28-day cultures (excluding *E. coli*).

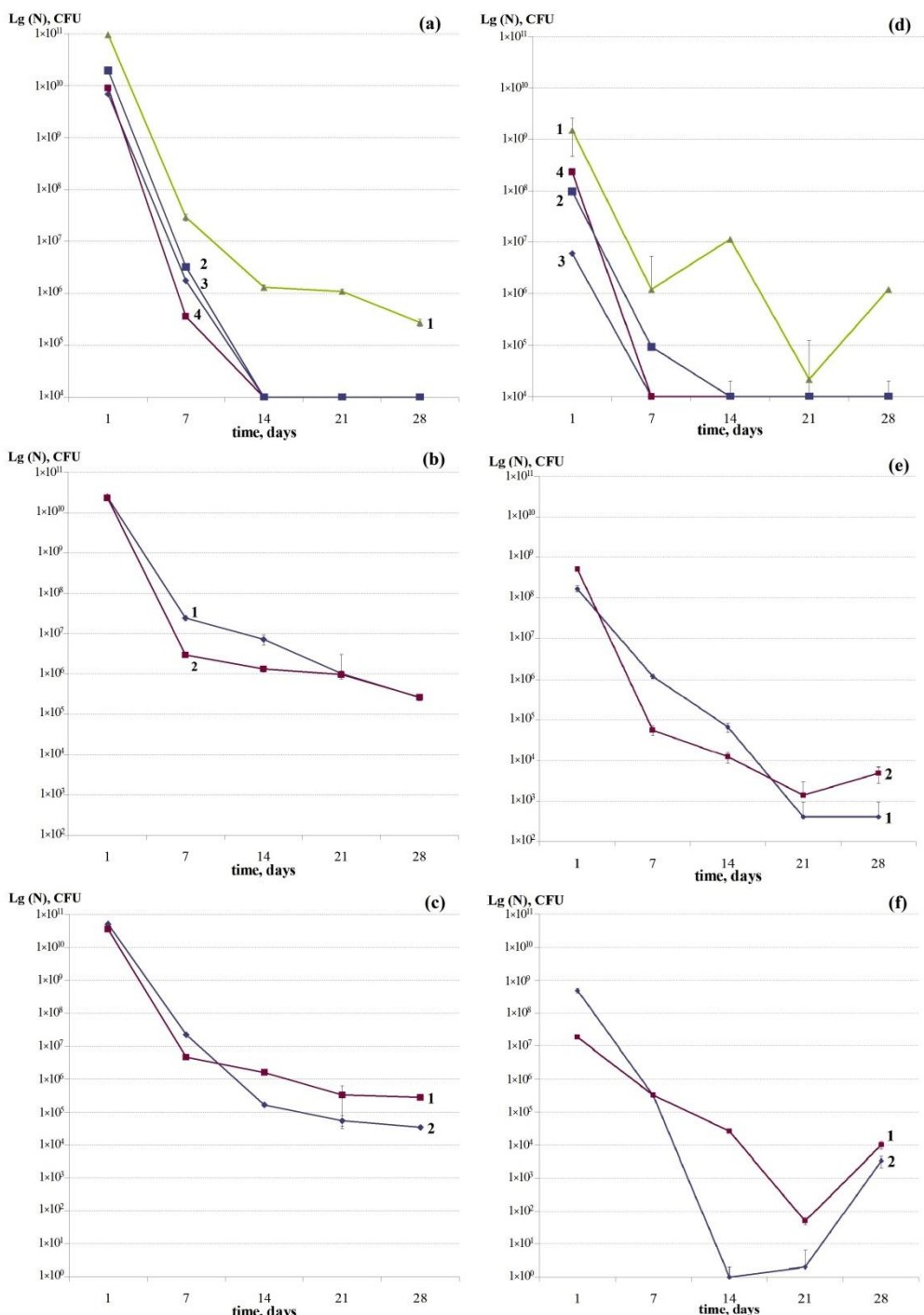

**Figure 1.** Dynamics of changes in the total number (CFU ml$^{-1}$) of cells (TNC) (**a**–**c**) and the number of thermoresistant cells (TRCs) (**d**–**f**) in the periodic planktonic bacterial cultures: (**a**,**d**)—*P. aeruginosa* (1), *P. extremorientalis* (2), *P. antarctica* (3); *P. gessardii* (4), (**b**,**e**)—*E. coli* (1), *S. typhimurium* (2); (**c**,**f**)—*S. aureus* (1), *K. rhizophila* (2).

**Table 1.** Dynamics of changes in the proportion (%) of thermoresistant cells relative to the total number of cells (CFU) in a sample before heating (100%) (in planktonic cultures with agitation).

| Strain | 18 h | 7 Day | 14 Day | 21 Day | 28 Day |
|---|---|---|---|---|---|
| *Pseudomonas aeruginosa* S481 | 1.56 | 4.11 | 863.64 | 1.96 | 437.96 |
| *Pseudomonas extremorientalis* S85 | 0.49 | 2.90 | 0.00 | 0.00 | 0.00 |
| *Escherichia coli* K12 | 0.69 | 4.88 | 0.92 | 0.04 | 0.16 |
| *Salmonella typhimurium* TA 1535 | 2.17 | 1.89 | 0.92 | 0.15 | 1.83 |
| *Kocuria rhizophila* S155 | 0.95 | 1.44 | 0.00 | 0.00 | 10.00 |
| *Staphylococcus aureus* ATCC6538 | 0.05 | 7.15 | 1.67 | 0.02 | 3.72 |

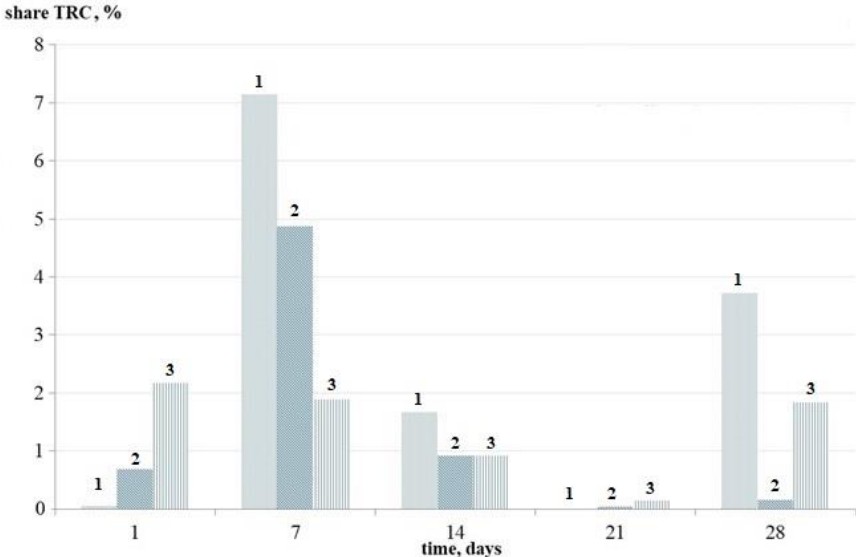

**Figure 2.** Dynamics of changes in the TRC proportion (%) in the developing periodic planktonic bacterial cultures: 1, *S. aureus*; 2, *E. coli*; 3, *S. typhimurium*.

The revealed peculiarities of the TRC dynamics in the cultures of all studied bacteria can be explained only by the cardinal change in the structure of cells that survived until the 28th day, namely, by vitrification of cytoplasm and its transition to a "glass-like" state due to the critical reduction of the amount of free water in a cell [47]. Based on the analysis of the obtained data, it can be suggested that cytoplasm vitrification in surviving cells takes place in the period between the 21st and 28th days of incubation, which explains an increase in the TRC number on the 28th day of incubation, when cytoplasm vitrification ends the process of maturation of surviving persister cells into dormant forms (DFs).

### 3.2. Dynamics of the Development of Mono-Species Biofilms of Bacteria under Study: Changes in the Number of Viable Cells, Antibiotic Tolerant Cells, and Thermoresistant Cells

#### 3.2.1. Dynamics of the Total Number of Cells of the Studied Bacteria

Effectiveness of biofilm formation by pseudomonades on fiberglass filters depended on species. Biofilm cultures of *P. antarctica* and *P. gessardii* were distinguished by the lowest survival rate (Figure 3) compared to other studied pseudomonades (Figure 4a). Only in the first seven days of incubation did the number of CFUs in formed biofilms of pseudomonades remain at a level of $10^9$ CFU (hereinafter, the number of CFU on a filter area of 225 mm$^2$) (Figure 3). During the following 12–28 days of incubation, the number of cells that were able to form colonies fell to zero (data are not presented). With that, persister (P) cells surviving in the presence of the lethal doses of antibiotics and TRCs were revealed only in the two-day biofilms. Their number was low (about $10^2$ CFU), which in combination with the data about the rapid loss of the ability to form colonies,

suggests that both pseudomonades are incidental contaminants in food enterprises and are not hazardous.

Other pseudomonades isolated as contaminants of food enterprises, *P. aeruginosa* and *P. extremorientalis*, turned to be more adapted to growth and survival in biofilms (Figure 4a).

In the mono-species BF of *P. aeruginosa*, the total number of cells (TNC) remained at a level of $\times 10^8$–$10^9$ CFU during the first 7–14 days of growth; after that, it began to decrease sharply and stabilized on the 14th day of incubation.

Studying the TNC dynamics during the biofilm growth of enterobacteria *E. coli* and *S. typhimurium* revealed their identity: a reduction in TNC by more than three orders of magnitude during two weeks with the following stabilization at a level of $10^5$ CFU (Figure 4b). Therefore, upon biofilm growth of the phenotype of two enterobacteria populations, a phenomenon of self-maintenance of the stable numbers of colony forming cells in an old biofilm was observed.

Gram-positive cocci *S. aureus* and *K. rhizophila* demonstrated the similar dynamics of the TNC development upon biofilm growth (Figure 4c). The TNC of *Staphylococcus* decreased at a higher rate than TNC of *K. rhizophila* up to the 14th day of incubation, reaching a plateau after the 14th day, while the number of CFU of *Kocuria* continued to decrease.

Therefore, during the growth of the bacteria under study in mono-species BFs, the stabilization of the total number of viable cells within $10^5$ CFU/filter occurred on the 14th day. It is unlikely that this stabilization is conditioned by the secondary growth of the population on the products of autolysis of biofilm cells, as it was found only after two weeks of incubation and was stably maintained up to the 25th day of observations. Based on the literature data and our own experimental experience, it is possible to suggest that in the developed biofilm cultures, the process of autolysis of ordinary vegetative stationary cells is coming to an end after two weeks and the developed plateau is represented by persister cells.

### 3.2.2. Dynamics of the Number of Antibiotic Tolerant Persister Cells Formed in Mono-Species Biofilms

The dynamics of changes in the number of persister cells of pseudomonades (Figure 4d) were characterized by a relative stability upon a somewhat increase in the number of P cells on the seventh day of the BF development. The number of P cells in *P. extremorientalis* was $10^2$ CFU/filter, and *P. aeruginosa* $10^3$ CFU/filter.

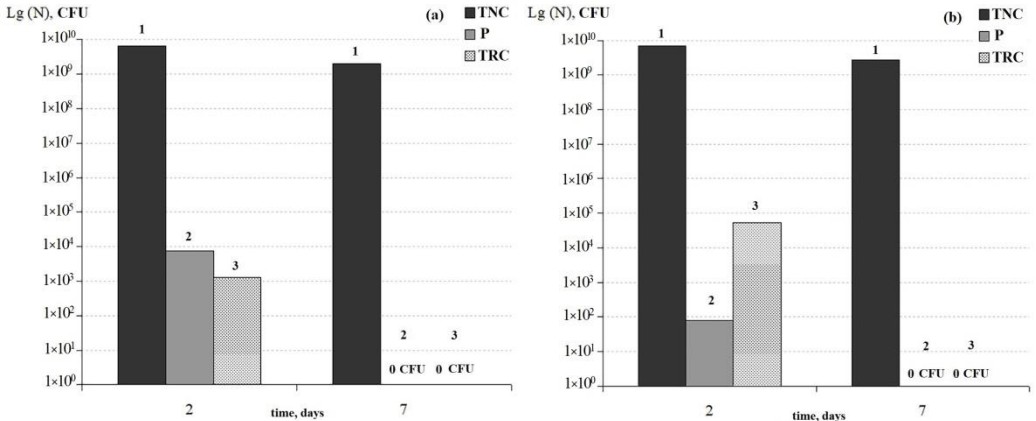

**Figure 3.** Changes in the total number (CFU/filter) of cells (TNC) (1), number of persisters (P) (2) and thermoresistant cells (TRC) (3) in the two-day and seven-day mono-species biofilms of *P. antarctica* (**a**) and *P. extremorientalis* (**b**).

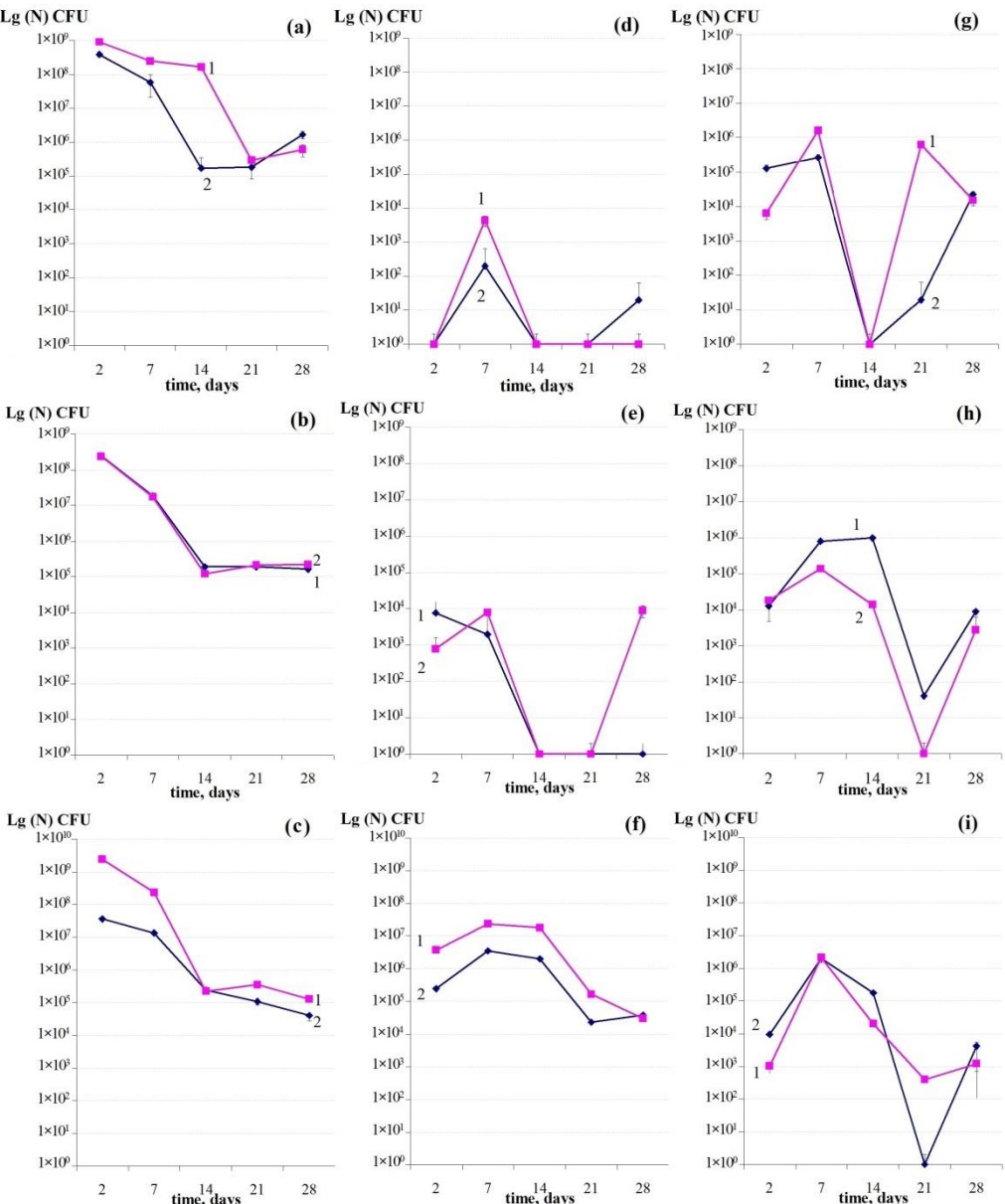

**Figure 4.** Dynamics of changes in the total number (CFU/filter) of cells (**a**–**c**), number of persisters (**d**–**f**) and thermoresistant cells (**g**–**i**) in the mono-species biofilms: 1, *P. aeruginosa* and 2, *P. extremorientalis* (**a**,**d**,**g**); 1, *E. coli* and 2, *S. typhimurium* (**b**,**e**,**h**); 1, *S. aureus* and 2, *K. rhizophila* (**c**,**f**,**i**) during 28-day incubation.

In the two- and seven-day biofilms of *E. coli*, the number of cells surviving after treatment with ciprofloxacin was about $10^4$ CFU/filter and that of *S. typhimurium* was about $10^3$ CFU/filter (Figure 4e). On the 14th–21st days of growth, the number of ATCs reduced to $10^2$ CFU in both bacteria and remained at this level up to the end of the experiment in the *E. coli* BF and increased up to $10^4$ CFU/filter in the *S. typhimurium* BF by the 28th day of growth. An increase in the number of P cells on the 28th day in *S. typhimurium*, apparently, can be explained by the transition of surviving P cells into the dormant state (DFs), which is characterized by stress resistance, including to antibiotics, as P cells are less stress-resistant than DFs.

The dynamics of changes in the number of P cells in BFs of gram-positive cocci differed from the dynamics of P cells in BFs of gram-negative bacteria. The number of P cells surviving in the presence of chloramphenicol (CP) increased up to the 14th day of BF growth and stabilized in the period of 21–28 days (Figure 4f). A higher proportion

of persisters (Supplementary Figure S1) in planktonic cultures of gram-positive bacteria compared to gram-negative bacteria was noted earlier [48,49] and possibly is linked with the structure of their cell walls. In the biofilm culture, *S. aureus* formed a slightly higher number of P cells than *K. rhizophila* (Figure 4f), which corresponds to the dynamics of the total cell number in their biofilm populations (Figure 4d).

It should be noted that, firstly, despite some differences, the dynamics of the revealed antibiotic tolerant P cells in gram-negative and gram-positive bacteria were principally the same: maximum P cells in the two- and seven-day biofilms with the following reduction in their numbers on the 21st day of incubation. Secondly, the number of P cells surviving in the presence of antibiotics was higher by two orders of magnitude in gram-positive bacteria than in gram-negative bacteria (Figure 4d,e). It is worth noting that the proportion of persisters in planktonic stationary cultures of gram-negative bacteria is up to 1% [29,32,34]. In the biofilms of gram-positive bacteria, the proportion of P cells corresponded to their proportion in the planktonic cultures and was from 0.3%–0.8% in the two-day BFs to 10%–12% in the seven-day BFs (Supplementary Figure S1). With that, in BFs of gram-negative bacteria, the proportion of P cells was significantly lower and accounted for 0.05% to 0.0005% (Supplementary Figure S1) on the 2nd day of growth, when the total number of cells was highest (Figure 4). Thirdly, the number of revealed antibiotic tolerant cells reduced on the 14, 21–28 days of biofilm growth and was $10^2$ CFU/filter in gram-negative bacteria and $10^4$ CFU/filter in gram-positive bacteria. This stabilization of the number of AT cells (Figure 4d–f) is consistent with the stabilization of the total number of cells in the 21-day and 28-day biofilms (Figure 4a–c). The revealed decrease in the number of persisters in all studied bacteria on the 14th–21st days can be explained by their transition to the viable but non-culturable (VBNC) state or dying. If P cells are not dead but have matured into DFs or temporally transformed to the VBNC state, heating (80 °C, 15 min) can facilitate their reactivation. The dynamics of bacterial biofilm cells under study that survived upon heating (80 °C, 15 min) was also investigated.

3.2.3. Dynamics of the Number of Thermoresistant Cells in Mono-Species Bacterial Biofilms

Analysis of the dynamics of thermoresistant cells (TRC) in biofilms developing over 21 days revealed its principal similarity for all studied cultures (Figure 4g–i). In the graphs of the TRC dynamics, their maximum is noted on the seventh day of the BF growth, which corresponds to changes in the number of AT cells (Figure 4d–f). It is most interesting that a sharp reduction in the TRC number was observed on the 21st day of biofilm incubation for BFs of all studied bacteria (for *P. aeruginosa* on the 14th day of growth) with the following likewise sharp increase in their number on the 28th day of incubation. Calculation of the proportion (%) of TRC relative to the total number of cells in the surviving populations of aging biofilms (Supplementary Figure S2) revealed regularities that were common for all bacteria under study: a high proportion of TRCs in the seven-day biofilms in a range of 0.05%–1% to 7%–10%, subsequent reduction (to 0%) on the 14th–21st days, and an increase in the proportion of TRCs in the 28-day biofilms up to the values typical of the seven-day biofilms. These results suggest that a subpopulation of persisters possessing not only antibiotic tolerance but also resistance to heat shock is formed in the bacterial biofilm populations on the seventh day of the BF development. This subpopulation persists throughout BF development, but the properties of thermal resistance in the 7-day persisters and 21-day DFs matured from persisters differ by the mechanism of maintenance of this state. It is necessary to note that the property of thermal resistance is considered a determining trait of anabiotic dormant (resting) forms (DFs), such as endospores, exospores, and cyst-like dormant cells (CLDCs) [25,27,45,46]. This property of DFs is determined by vitrification of their cytoplasm at the last stage of their maturation. Persister cells also possess thermal resistance, but their resistance mechanism is different [34,50]. The selection of spontaneous mutations that enable growth of antibiotic- and heat-resistant mutants was excluded logically. The possibility for any specific mutant is $10^{-8-9}$ per dividing

cell/generation. Meanwhile, $10^6$ cells were applied to each glass filter. Hence, content of possible mutants (heat-tolerant or ABR) in the grown biofilm population is negligible. The GASP (growth advantage in stationary phase) mutants could be excluded since glass filters were seeded by cells of the early stationary phase, not of the late stationary phase [51].

An agreement between the dynamics of the number of P cells and TRCs in developing biofilms allows us to suggest that during the long-term incubation of persisters, a further reduction of their weak metabolic activity takes place. At this stage of P cell survival, the exposure to antibiotics or heat shock leads to a loss of the ability to form CFU (on the 21st day). During further incubation, however, the process of cytoplasm vitrification occurs in persisters, which is the main mechanism of stress resistance, including thermal resistance of matured DFs. The obtained results confirm the authors' hypothesis that P cells are precursors of DFs, in which molecular–genetic processes have ended, but their transition into the anabiotic state has not. Based on the obtained results, it can be suggested that vitrification in persisters occurs in a period between 21–28 days of incubation.

The formulated conclusions are confirmed by the correlation analysis of indicators of P cell numbers and TR cell numbers (Table 2). A weak positive correlation for TRCs/TNC, that is, the direct dependence of the number of TR cells on their total number, was observed for *P. extremorientalis* (0.37), which indicates the weak relationship between these indicators. In other cases, correlation was negative, which points to an increase in the proportion of TRCs upon a decrease in the total number of cells (TNC) in the population. This conclusion agrees with the results of the correlation analysis of the data array for TNC and the number of persisters. A strong positive relationship between TNC and the number of P cells was revealed for *E. coli* (correlation coefficient 0.98) and for *K. rhizophila* (correlation coefficient 0.2).

**Table 2.** Values of correlation coefficients between indicators of total number of cells (TNC), number of thermoresistant cells (TRCs), and persister (P) cells.

| Test Culture | Correlation Coefficient (Cp) | | |
|---|---|---|---|
| Strain | TNC/TRC * | TNC/P | TRC/P |
| *Pseudomonas aeruginosa* S481 | −0.19 | −0.08 | 0.92 |
| *Pseudomonas extremorientalis* S85 | 0.37 | −0.10 | 0.88 |
| *Escherichia coli* K12 | −0.32 | 0.98 | −0.28 |
| *Salmonella typhimurium* TA 1535 | −0.10 | −0.33 | 0.50 |
| *Staphylococcus aureus* ATCC6538 | −0.18 | −0.17 | 0.75 |
| *Kocuria rhizophila* S155 | −0.02 | 0.19 | 0.88 |

* TNC—total number of cells; TRC—thermoresistant cells; P—persister cells.

A different situation is observed for correlation of the number of TR cells and persisters (on the 28th day—DFs). For all biofilm cultures under study, a positive relationship between these indicators was shown; that is, an increase in the proportion of persisters was parallel with an increase in the proportion of TR cells in populations. The exclusion was *E. coli*, which showed a negative correlation coefficient (a weak negative relationship with a coefficient of −0.3). A strong relationship between the development of P cells and TR cells confirms the earlier proposed hypothesis that regards P cells as precursors of dormant forms (DFs) that are formed on the 28th day of biofilm incubation and are resistant to thermal treatment.

The obtained results suggest that the studied microorganisms, including contaminants of food production, realize similar survival strategies during their growth in biofilms; however, the degree of their reaction to substrate exhaustion, exposure to antibiotics, and high temperatures depends on their taxonomic status. The gram-positive bacteria and members of *Enterobacteriaceae* (*E. coli* and *S. typhimurium*) turned out to be most resistant compared to pseudomonades. The phenomenon of formation of dormant cells (which are

stable to non-vital temperatures) from persisters has a multi-level character and, apparently, is related to a physiological state of surviving persister cells at a given moment.

### 3.3. Comparison of the Dynamics of Thermoresistant Cell Formation in Planktonic and Biofilm Cultures

Comparison of the dynamics of the thermoresistant cell formation during long-term incubation (28 days) of the studied biofilm and planktonic bacterial cultures revealed their principal identity as well as their common peculiarity: a decrease in the number of TR cells on the 21st day of incubation and a sharp increase on the 28th day (Figures 5 and 6) according to the above explanation about the manifestation of thermal resistance in persisters and anabiotic dormant forms (DFs) maturing from them. With that, the growth type of bacteria under study (biofilm or planktonic) did not significantly affect the number of TR cells in a unit of area (in planktonic cultures—CFU/mL of medium, in biofilms—CFU/filter, 225 mm$^2$). The number of TR cells depended more on the bacterial taxonomic affiliation (Figure 5). However, the proportion (%) of TR cells in the developing bacterial cultures was significantly influenced by their growth type (planktonic of biofilm) (Figure 6). The proportion of TR cells was higher in the planktonic cultures of *S. typhimurium*, *P. aeruginosa*, and *S. aureus*, while in *E. coli*, it was higher upon biofilm growth. It is possible that this fact reflects the high adaptation of *E. coli* to growth as a biofilm phenotype typical of this bacterium, whose main ecological niche is the gut. Detection of *E. coli* in the biofilm composition in industrial enterprises is an alarming signal.

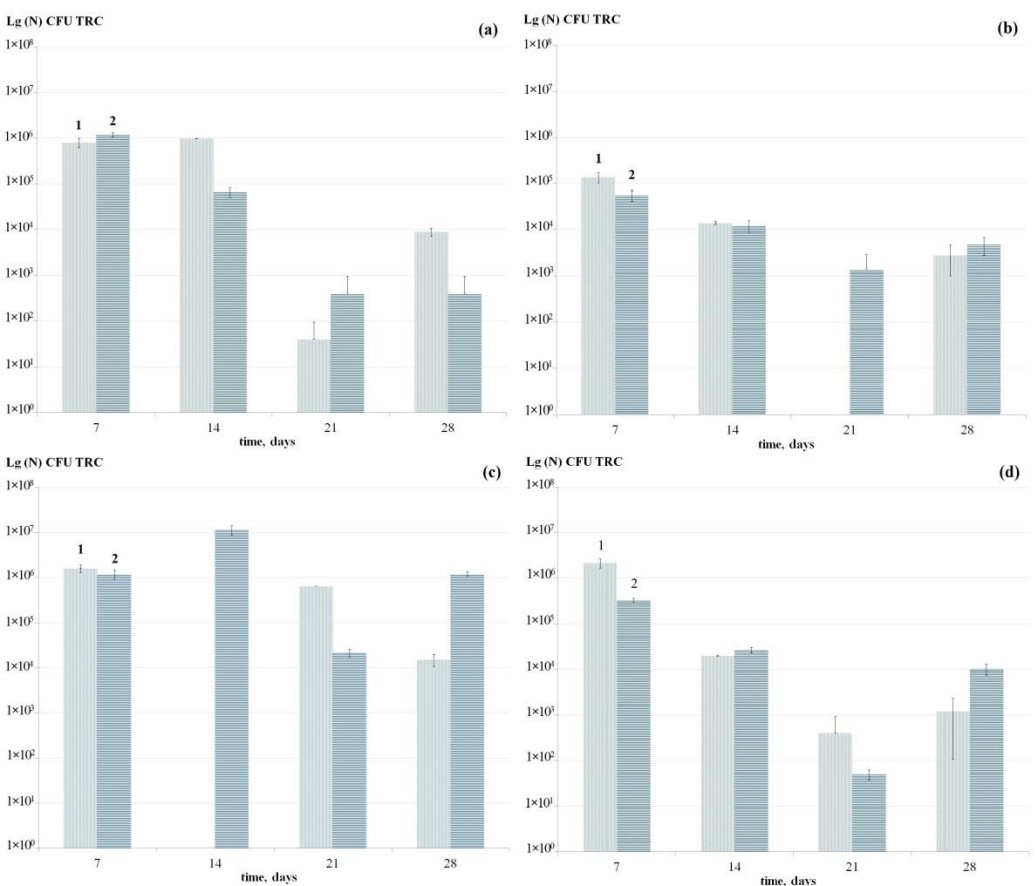

**Figure 5.** Dynamics of changes in the number (CFU/filter) of thermoresistant cells (TRC) in the biofilms (1) and planktonic cultures (2) of bacteria: *E. coli* (**a**), *S. typhimurium* (**b**), *P. aeruginosa* (**c**), *S. aureus* (**d**) during 28-day incubation.

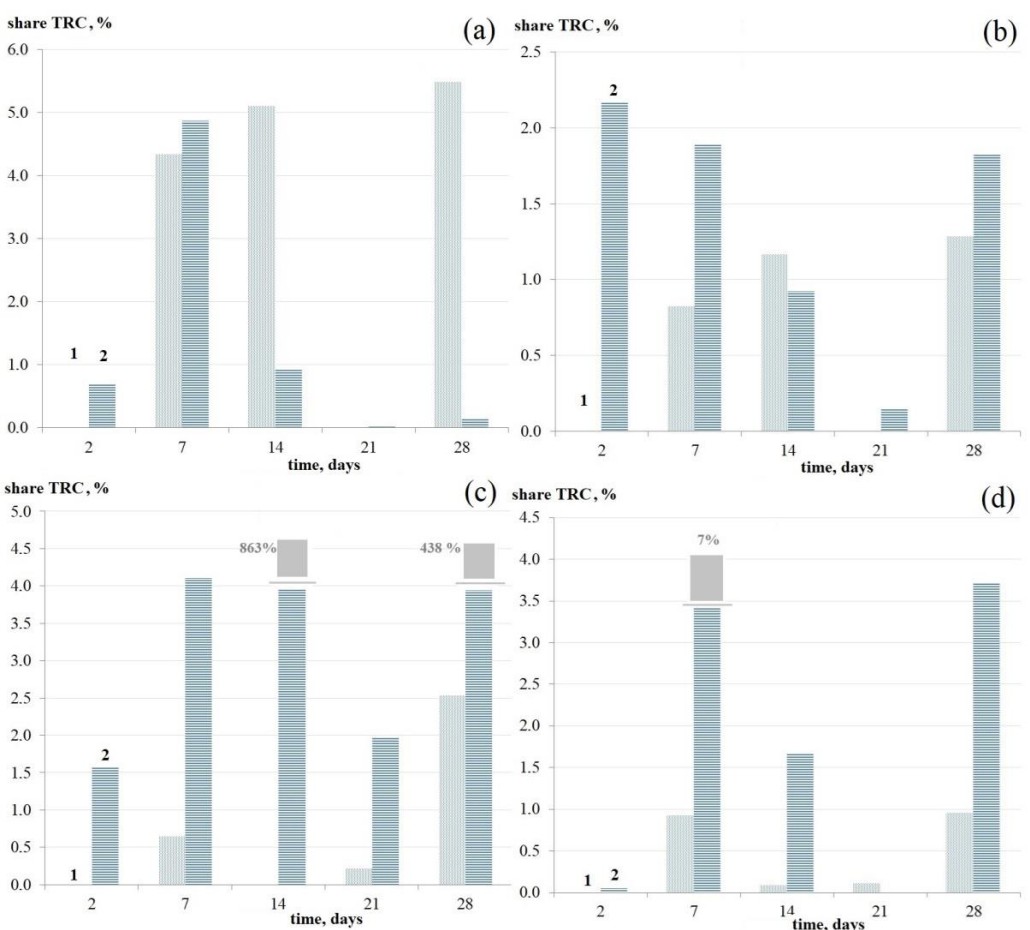

**Figure 6.** Dynamics of changes in the proportion (%) of thermoresistant cells in the biofilms (1) and planktonic cultures (2) of bacteria: *E. coli* (**a**), *S. typhimurium* (**b**), *P. aeruginosa* (**c**) and *S. aureus* (**d**) during 28-day incubation.

### 3.4. Binary Biofilm Development

Binary biofilms were obtained on the fiberglass filters (225 mm$^2$) by inoculation of aliquots (20 μL each) of stationary submerged bacterial cultures mixed beforehand (in equal volumes of 500 μL each). The bacterial pairs were as follows: *E. coli* + *K. rhizophila* and *S. typhimurium* + *S. aureus*. Biofilms were incubated at 28 °C. The most accurate determining parameter that reflected the bacterial development in binary biofilms was the number of antibiotic tolerant persister cells, which was detected by survival of gram-negative bacteria in the presence of ciprofloxacin and the gram-positive bacteria in the presence of chloramphenicol. Over a period of two to seven days of incubation, the number of cells of both cultures increased; with that, a higher increase was observed in *K. rhizophila* than in *E. coli* (Supplementary Figure S3). The proportion of P cells in the seven-day binary BFs reduced in both bacteria, but the reduction was higher in *E. coli*; that is, it was disproportional relative to the growth in the total number of cells (TNC).

Another binary BF (*S. typhimurium* and *S. aureus*) was characterized by a significantly lower proportion of persisters of both biofilm cultures on the second day of incubation (Supplementary Figure S4). With that, the number of P cells of *S. typhimurium* did not change in the seven-day BFs, and the proportion decreased, while in *S. aureus* in the seven-day BFs, both the number of P cells and their proportion increased.

In both binary BFs, the number of P cells on the seventh day was significantly (by 4–5 orders of magnitude) higher than in mono-species BFs (Figure 4). This phenomenon was noticed for the first time (methodologically, the number of P cells in mono- and binary BFs was measured identically). The biofilm phenotype in both cases was also

manifested identically, as can be concluded from the electron microscopic investigations (see below). Consequently, the noticed increase in the number of P cells in binary BFs can be explained only by interactions between cells of different bacterial populations in BFs. These results agree with a lot of previously obtained data that explain the higher stress resistance of multi-species BFs compared to mono-species BFs by interactions between biofilm populations [6,7,17–20]. Our results correspond to the theory of P cell formation, where one of the mechanisms of this process is induction of the quorum sensing (QS) system by auto-regulators [32,41,52]. In our experiments, gram-positive bacteria showed a significant advantage in formation of stress-resistant P cells in both binary BFs (Table 3).

**Table 3.** Values of the persister (P) proportions (%) relative to the total number of cells (100%) in the mono-species and binary biofilms.

| Variants of Combinations | P, %, 7th Day | P, %, 14th Day |
|---|---|---|
| Mono-biofilm *E. coli* K12 | 0.01 | 0 |
| Binary biofilm *E. coli* K12 | 0.46 | 0.1 |
| Mono-biofilm *K. rhizophila* S155 | 25.97 | 83.47 |
| Binary biofilm *K. rhizophila* S155 | 8.15 | 4.76 |
| Mono-biofilm *S. typhimurium* TA1535 | 0.05 | 0 |
| Binary biofilm *S. typhimurium* TA1535 | 0.1 | 0 |
| Mono-biofilm *S. aureus* ATCC6538 | 10.25 | 81.08 |
| Binary biofilm *S. aureus* ATCC6538 | 3.67 | 20.42 |

*3.5. Electron Microscopic Studies of Mono-Species Biofilms*

Whereas survival of the population is ensured by formation of stress-resistant P cells in the developing planktonic cultures, in old cultures (one month and more), it is ensured by formation of anabiotic dormant forms (DFs). The determining characteristics of DFs, besides their high stress resistance, in particular, thermal resistance, are distinctive features of the ultrastructural organization: thickened cell wall and bio-crystallized nucleotide developing as a result of the interaction of the stationary phase protein (Dps) with DNA, which leads to the development of the condensed nucleotide in the central part of DFs. When studying biofilms isolated in meat processing plants, we systematically found DFs in them, which were well-diagnosed using electron microscopy of thin sections of DFs [42]. In the present work, we studied the organization of the 6-day mono-species BFs (Figure 7) and cell ultrastructure of 2- and 28-day BFs of the model bacteria (Figure 8).

Scanning microscopy of the six-day BFs of *K. rhizophila* (Figure 7a,b) revealed a well-developed BF with a large array of cells that were tightly adherent to each other (Figure 7a), and multiple cells adhered on filter fibers (Figure 7b). The six-day biofilm of *S. typhimurium* was also characterized by aggregation of cells and a well-developed matrix with immersed cells adhering to the fiberglass (Figure 7c,d). It is necessary to note that *K. rhizophila* cells do not grow into the depth of the filter and form a BF mainly on its surface, while the *Salmonella* BF develops in the filter depth. The depth of penetration and the character of cell–substrate adhesion allow us to suggest that the *Kocuria* BF is easier to wash from the surface than the *S. typhimurium* BF.

Transmission electron microscopy of sections of 2-day and 28-day BFs of model bacteria revealed significant differences in their ultrathin organization (Figure 8).

For two-day biofilm bacteria *E. coli* (Figure 8a,b) and *P. aeruginosa* (Figure 8d,e), which represent stationary cells, the following distinctive characteristics were observed:

(1) structure of the cell envelope typical of gram-negative bacteria; (2) formation of granules of bacterial storage material—poly-hydroxybutyric acid (PHBA); (3) formation of multiple vesicles typical of the biofilm phenotype that contain hydrolytic enzymes, which facilitate utilization of the substrate and are attached to the outer membrane and also are detaching from it [42]; (4) the beginning of the development of the bio-crystal nucleotide is well seen in the two-day biofilm stationary cells (Figure 8b–e), which was described earlier as the cloddy texture of the cytoplasm.

Two-day biofilm bacterium *K. rhizophila* (Figure 8g,h) showed the thickened cell wall typical of gram-positive bacteria, the absence of poly-hydroxybutyric acid (PHBA), development of vesicles, and the beginning of formation of bio-crystal nucleotides.

In the 28-day BFs of both gram-negative and gram-positive bacteria, bacterial dormant forms (DFs) that differed structurally from vegetative cells were well diagnosed. The DFs of the bacteria under study demonstrated thickening of cell envelopes, especially pronounced in DFs of gram-positive bacteria (Figure 8i), absence of vesicles as one of the signs of the metabolically inactive state, and bio-crystal nucleotides filling the whole volume of the cell with the centrally located zone of the compacted nucleotide (Figure 8c,f,i).

Therefore, the development of biofilm cultures of both non-spore-forming gram-positive and gram-negative bacteria ends with the development of dormant forms (DFs) similar to DFs formed in the periodic planktonic bacterial cultures in terms of the ultra-structural organization [25,27,53].

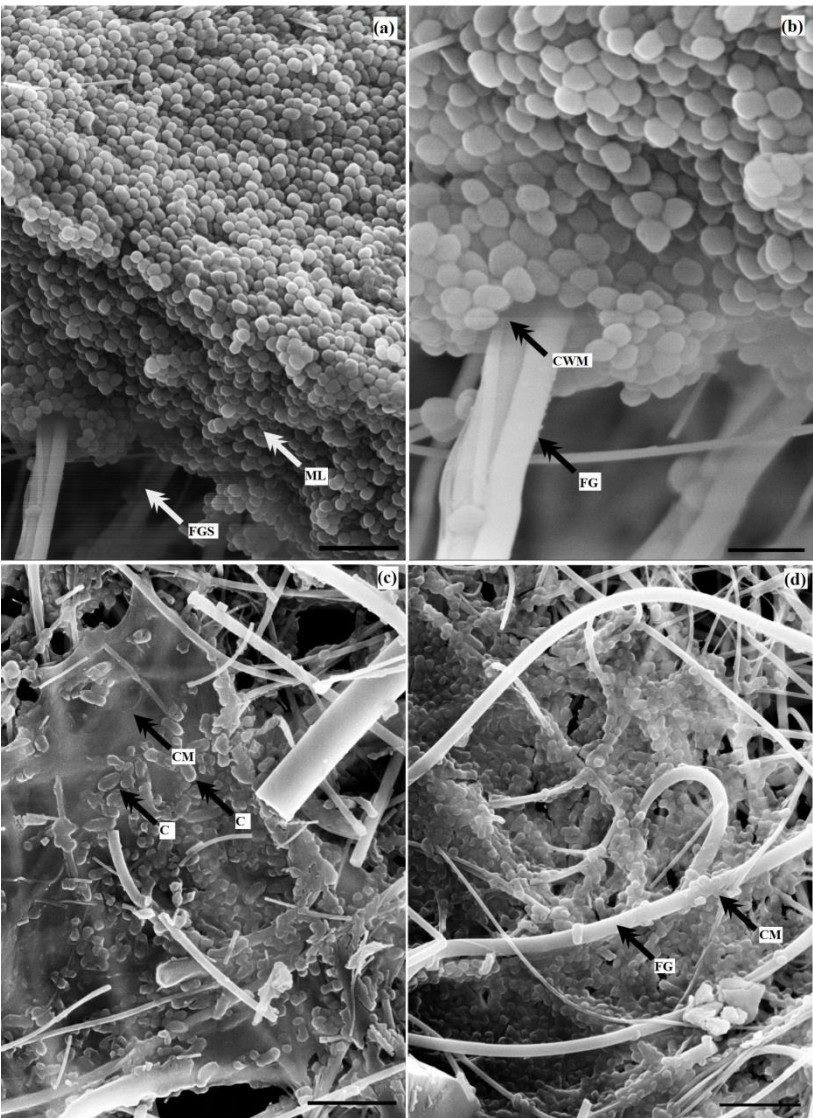

**Figure 7.** Overall appearance of the mono-species biofilms of *K. rhizophila* (**a,b**) and *S. typhimurium* (**c,d**) on the sixth day of incubation. Scanning electron microscopy (SEM). FGS, fiberglass substrate; ML, multicellular layer; CWM, cells without matrix; FG, fiberglass; C, cells; CM, cellular matrix. Bars, 5 μm (**a,c,d**); 2 μm (**b**).

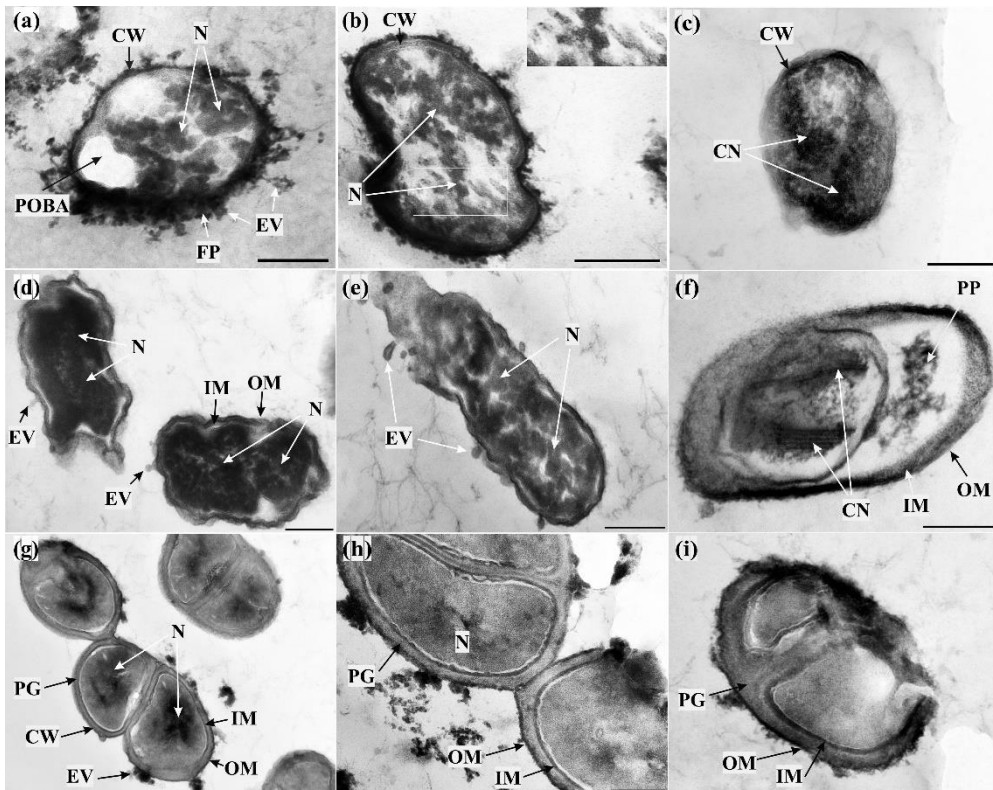

**Figure 8.** Cell morphology in *E. coli* (**a–c**), *P. aeruginosa* (**d–f**) and *K. rhizophila* (**g–i**) on the 2nd (**a,b,d,e,g,h**) and 28th (**c,f,i**) days of incubation in the mono-species biofilms. N, nucleotide; CN, crystallized nucleotide; CW, cell wall; OM, outer membrane; IM, inner membrane; PP, periplasm; PG, peptidoglycan; POBA, poly-oxybutyric acid; EV, extracellular vesicles; FP, fibrillar polysaccharide. Bar, 0.2 μm.

## 4. Discussion

Both in medicine and in biotechnology, the main direction for studying BFs (mainly, of pathogenic bacteria) is elucidation of the causes of their high stress resistance determining their survival and spread in a specific environment. High levels of bacterial survival in BFs have been linked, first of all, with the structural organization of BFs and the presence of the polymer matrix, complex intercellular interactions in multi-species BFs, and recently with the physiological age-related heterogeneity of biofilm bacteria [12,21,22]. Studies of the phenotypical heterogeneity of the biofilm population, namely, the development of the subpopulation of stress-resistant persister cells and DFs as one of the important causes of high survival of BFs upon lethal exposure, are presented poorly. It should be noted that comprehensive studies of P cells as the main phenotype of antibiotic tolerance, development of antibiotic resistance, and the cause of recurrent infections are carried out on planktonic periodic bacterial cultures; however, biofilms are the main form of their existence.

In the present work, the comparative study of the development of bacterial populations and generation of forms of their survival upon the planktonic and biofilm growth of non-spore-forming bacteria, including those that were isolated from BFs taken in meat processing enterprises, was carried out.

An emphasis in the study was made on formation of the subpopulation of P cells as the main phenotype of population survival in the presence of lethal doses of antibiotics, disinfectants, and other stress impacts, as well as anabiotic dormant forms that are stress-resistant, in particular, thermoresistant, and ensure survival of the population in conditions not favorable for growth. It is necessary to note that traditionally, antibiotic tolerance is considered the main property of P cells; however, in the natural systems, lethal concentrations of antibiotics are not found. Therefore, when modeling and studying mono-species

and binary BFs, two parameters were used as an indicator of stress resistance of P cells and DFs: (1) antibiotic tolerance—survival at a dose of antibiotics of 10 MIC, and (2) thermal resistance—survival at heat shock (heating at 80 °C, 15 min); that is, in conditions where vegetative bacterial cells die. Note that suspensions of biofilm bacteria and not BFs per se were exposed to stress, which enables judging about properties of cells not touching an effect of the matrix (see Materials and Methods).

Analysis of the obtained results allows us to make the following main conclusions.

(1) The first conclusion follows from the results, showing that practically the same number of cells (~$10^5$ CFU/mL or CFU/filter) survive in old 28-day cultures (both biofilm and planktonic); among them, approximately the same number of cells (~$10^4$ CFU/mL or CFU/filter) also possesses stress resistance (thermal resistance) (Figures 1 and 4). Therefore, the number of surviving stress-resistant cells in bacterial populations that develop and die in the same standard conditions (medium, temperature) poorly depends on their taxonomic status and phenotype of the development (planktonic or biofilm). In other words, upon the periodic development of bacterial populations, approximately the same number of surviving stress-resistant cells per unit of space (volume) is formed in them. The first example of "spatial" regulation of the development of bacterial populations was regulation of their numbers in a volume—density regulation of QS systems. Our obtained results can be regarded as the second example of spatial regulation of the development of populations. (2) It has been shown experimentally that the proportion (%) of TR cells in planktonic and biofilm bacterial cultures during the late period of their development can exceed the total number of bacteria (both indicators are determined by plating into solid growth media), as for instance, in *P. aeruginosa* on the 28th day of its development (Figure 1a,d). The revealed inconsistency is explained by the fact that in old bacterial cultures, the forms of survival are anabiotic dormant cells (DFs), whose population is always heterogeneous in terms of DF properties such as their germination in standard conditions and thermal resistance [26,27]. Heating that we used (80 °C, 15 min) activated germination of dormant cells, including the subpopulation of DFs that are difficult to germinate, which led to the excess in their numbers. This method is known as activation of bacterial endospores and is used in biotechnological practice. Mechanisms of heat activation are an increase in the micro-fluidity of the lipid stroma of the plasma membrane [25]. (3) Analysis of the dynamics of AT cells surviving in the presence of antibiotics and reverting to growth in a fresh medium as well as TR cells resistant to heat shock (80 °C, 15 min.) revealed a regularity identical for biofilm and planktonic cultures. In aging populations, antibiotic resistant or thermoresistant cells were not found on the 21st or 14 day of incubation (Figures 1 and 4); however, their number sharply increased to $10^3$–$10^4$ CFU/mL (or CFU/filter) on the 28th day of incubation.

This phenomenon noticed for the first time can be explained only by acquisition of the non-culturable state by persisters incubated for a long time; that is, the loss of the ability to form colonies on solid media in the standard conditions. This state was temporary; the surviving cells restored the ability to form colonies on the 28th day of incubation. So, what was happening with cells of the 21-day and 28-day biofilm and planktonic cultures?

The question about mechanisms and forms of survival of bacterial populations in the presence of antibiotics and other antimicrobial substances (disinfectants) has been widely discussed today [35,37,54]; however, all agree that given a strong stress burden, cells with reduced or stopped metabolism (BRHM) are forms of bacterial survival [54–56]. These forms include P cells, anabiotic DFs, and viable but non-culturable (VBNC) cells. Note that there is no general agreement regarding the definition of VBNC today. We think that VBNC cells described by Colwell [57] and Kaprelyantz [58,59] and temporally non-culturable cells that were detected by us are forms with different states. Special debates have been aroused by the question about the fate of P cells upon long-term incubation of bacterial cultures, which has a direct relationship with bacterial survival in BFs and effectiveness of disinfection measures and antibiotic therapy. One group of researchers believes that upon long-term incubation of bacteria, the "dormant" metabolic state of P cells is deepened and

they are transformed to the non-culturable (VBNC) state [36]. Other researchers are of the opinion that the non-culturable (VBNC) state is a "consequence of stress accumulation" and P cells as a result of cytodifferentiation are not relevant to VBNC cell formation; upon long-term incubation, P cells die [60].

In the present study, during long-term incubation of both BFs (Figure 4) and planktonic cultures (Figure 1), we found a phenomenon not observed earlier: with maintenance of viability of a small subpopulation of cells throughout BF incubation (28 days), cells changed their properties dramatically on the 14th–21st days of survival; that is, they lost completely the properties of antibiotic tolerance and thermal resistance, but restored these properties on the 28th day of incubation. As conditions of incubation of BFs and planktonic cultures did not change, this phenomenon can be explained only by dramatic changes in "quality" of surviving cells. Recall that P cell resistance to damaging impacts is conditioned by metabolic processes of realization of stress reaction, closure of targets for antibiotics, changes in the structural organization of membranes, and work of the efflux systems [32,36,37]. It is significant that the process of nucleotide bio-crystallization takes place both in P cells and in stationary cells [61]; however, due to extremely low metabolism in P cells, it occurs very slowly. The complex of metabolic processes determines stress resistance of P cells, including to antibiotics and heat shock. However, this stress resistance reduces as the weak metabolic activity typical of P cells extinguishes. With that, the process of dehydration [45] begins in stationary cells, including P cells forming as a subpopulation of stationary cells [29,34]. It follows from the theory of anabiosis and practice of anabiotic cell generation that when an amount of free water in cells (aw) reduces up to the critical level (<58%), the process of cytoplasm vitrification [48,62,63], the condition characteristic of bacterial DFs [25], takes place simultaneously.

Analysis of data obtained in the present work allows us to suggest with great probability that when the metabolic level has been critically reduced and vitrification still has not begun in P cells of cultures incubated for a long time, additional stress, such as an exposure to antibiotics or heat shock, determines transition of P cells into the non-culturable (VBNC) state, which was noticed by us on the 21st day of incubation of BFs (Figure 4d–i) and planktonic cultures (Figure 1d–f). These considerations do not contradict the theory of VBNC cell formation as a result of "stress accumulation" [64] and, in our opinion, P cells lose their colony forming ability and convert into the VBNC state due to the reorganization of the cell structure—that is, global aggregation of proteins and dimerization of ribosomes [36,63].

Upon further BF incubation, P cells undergo the process of cytoplasm vitrification during the incubation period of 21–28 days and pass into an anabiotic state, which confirms the earlier proposed hypothesis that regards P cells as precursors of anabiotic DFs [63,64]. When plated in a fresh medium, DFs germinate and grow cells as well as cells grown from persisters that demonstrate sensitivity to antibiotics or heat shock (Figures 1 and 4). We think that these results have fundamental significance.

The practical significance of the obtained results resides in the following recommendation for increasing effectiveness of disinfection procedures in food industry enterprises. Treatment with hot steam of biofilms, especially aging ones, in which mainly P cells and DFs developed from them survive, will lead to an increase in the fluidity of the lipid stroma of membranes and melting of the bio-crystal nucleotide. Thereby, P cells and DFs will lose stress resistance that is typical of them and lose viability upon their subsequent (or simultaneous) exposure to disinfectants. This method can be effective both for old BFs in places that are seldom subjected to disinfection and for BFs formed in places that are systematically treated with disinfectants, as high temperatures increase membrane fluidity in cells of any physiological age. An enhancement of the anti-biofilm effect of disinfectants can also be expected when using chemical compounds of the membranotropic type of activity as adjuvants.

In general, the results obtained make it possible to correct approaches to assessment of viability and stress resistance of cells at different developmental stages of bacterial BFs

and to design strategies for increasing the effectiveness of various methods and means to prevent the development of bacterial BFs in food industry enterprises.

**Supplementary Materials:** The following are available online at https://www.mdpi.com/article/10.3390/coatings12121913/s1. Figure S1: Proportion (%) of persisters relative to the total number of cells (100%) (CFU/filter) in the mono-species bacterial biofilms on days 2, 7, 14, 21 and 28 of bacterial biofilm incubation: 1, *P. extremorientalis*; 2, *E. coli*; 3, *S. typhimurium*; 4, *K. rhizophila*; 5, *S. aureus*; 6, *P. aeruginosa*; Figure S2: Proportion (%) of thermoresistant cells relative to the total number of cells (100%) in the mono-species bacterial biofilms of six bacterial species on days 2, 7, 14, 21 and 28 of biofilm incubation: 1, *P. extremorientalis*; 2, *E. coli*; 3, *S. typhimurium*; 4, *K. rhizophila*; 5, *S. aureus*; 6, *P. aeruginosa*; Figure S3: Changes in the number and proportion of persisters in the binary biofilm of *E. coli* (1) and *K. rhizophila* (2) on the 2nd and 7th day of incubation; Figure S4: Changes in the number and proportion of persisters in the binary biofilm of *S. typhimurium* (1) and *S. aureus* (2) on the 2nd and 7th day of incubation.

**Author Contributions:** Conceptualization, Y.A.N. and T.A.P.; methodology, Y.A.N., Y.K.Y. and T.A.P.; validation, E.N.T. and G.I.E.-R.; formal analysis, T.A.P. and Y.A.N.; investigation, T.A.P., E.N.T. and G.I.E.-R.; resources, Y.A.N. and Y.K.Y.; data curation, T.A.P., E.N.T. and Y.A.N.; writing—original draft preparation, T.A.P., G.I.E.-R. and Y.A.N.; writing—review and editing, T.A.P., Y.A.N. and Y.K.Y.; visualization, T.A.P., Y.K.Y. and E.N.T.; supervision, Y.A.N. and Y.K.Y.; project administration, Y.A.N. and Y.K.Y.; funding acquisition, Y.A.N. and Y.K.Y.; All authors have read and agreed to the published version of the manuscript.

**Funding:** This work was funded by grant NO. 075-15-2020-775 of the Ministry of Education and Science (for priority directions of scientific and technological development).

**Institutional Review Board Statement:** Not applicable.

**Informed Consent Statement:** Not applicable.

**Data Availability Statement:** Not applicable.

**Acknowledgments:** Electron microscopic studies were performed using the equipment of the UNIQEM Microbial Collection Center at the Research Center for Biotechnology of the Russian Academy of Sciences.

**Conflicts of Interest:** The authors declare no conflict of interest.

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
