# Peer review of "Forms of Bacterial Survival in Model Biofilms"

_coatings, doi:10.3390/coatings12121913_

Round 1
Reviewer 1 Report
in this research, the authors studied the development of resistance forms in planktonic bacteria and biofilms after exposure to antibiotics and heat.
This study could be interesting, but it contains many issues to be addressed.
major issues:
The manuscript is difficult to read. Authors should thoroughly review the manuscript in the introduction, methods, and results.
The choice of bacteria is not clear. There is extensive literature on some bacterial species and authors should cite previous work.
In the methods, it is not clear which bacteria are grown in co-culture and why these bacteria were chosen.
The results are not clearly shown. it is necessary for the results to mention the bacteria and the values obtained. It is not possible to be generic
Author Response
The authors thank the reviewer for all the comments.
Question-1: The manuscript is difficult to read. Authors should thoroughly review the manuscript in the introduction, methods, and results.
Answer: This demand is not clear enough and is not specific, and therefore it is difficult to fulfil it.
We agree that the style of introduction and presentation of results is somewhat "heavy" and have tried, where possible, to eliminate "heavy" constructions.
Question-2: The choice of bacteria is not clear. There is extensive literature on some bacterial species and authors should cite previous work.
Answer: when selecting cultures in binary biofilms, we took into account the possibility of their cultural and morphological distinction on an agarized medium, similar growth rates of cultures in liquid media, and, based on the literature data, we took into account the competitive interactions of Gram-negative and Gram-positive bacteria in binary biofilms. It was necessary to avoid too strong competition. We tested few pairs and have chosen pairs described in the paper.
The article has references to papers where the main subjects are the bacteria we use, and we also cite literature discussing binary systems, such as,
- Sharahi, J.Y.; Azimi, T.; Shariati, A.; Safari, H.; Tehrani, M.K.; Hashemi, A. Advanced strategies for combating bacterial biofilms. Cell Physiol. 2019, 234, 14689–14708, doi: 10.1002/jcp.28225.
- Yuan, L.; Hansen, M.F.; Røder, H.L.; Wang, N.; Burmølle, M.; He, G. Mixed-species biofilms in the food industry: Current knowledge and novel control strategies. Rev. Food Sci. Nutr. 2020, 60(13), 2277-2293, doi: 10.1080/10408398.2019.1632790.
Question-3: In the methods, it is not clear which bacteria are grown in co-culture and why these bacteria were chosen.
In section 2.3 we added information about the composition of binary biofilms. We added two suggestions to paragraph 2.4. (Determination of the number of viable cells in biofilms) that clarify the methodology of counting colonies of different species on agar plates.
Question-4: The results are not clearly shown. It is necessary for the results to mention the bacteria and the values obtained. It is not possible to be generic.
Dear Reviewer! Your demand is not clear to us. All our colleagues understood paper normally. We can not improve it anymore. We believe that few excessive numbers and names will make it more “heavy”, as all information is given in figures and tables.

Reviewer 2 Report
In the presented manuscript Pankratov et al investigated the formation of antibiotic tolerant (P, persisters) cells and anabiotic dormant forms (DFs) in planktonic populations and biofilms exposed to antibiotics and heat stress. The authors found the during the growth of the various Gram-negative and Gram-positive strains the number of thermo- and antibiotic-tolerant (P) cells initially decreased, and then, on the 28th day of growth, unexpectedly increased.Electron microscopy revealed the presence of DFs in the late 28-day biofilms. The Authors concluded that cytoplasm vitrification and other changes in cell ultrastructure were responsible for the transformation of P cells into DFs. Since heat stress can restore DFs growth, enhanced levels of surviving cells were observed at the end of the experiments. I think that to confirm these conclusions, another scenario should be excluded - the selection of spontaneous mutations that enable growth of antibiotic- and heat-resistant mutants or, in general, GASP (growth advantage in stationary phase) mutants. For example, GASP mutants appear in 10-day-old E. coli cultures (see DOI: 10.1126/science.7681219).
It is not clear how the number of persisters was estimated. According to ref. [41], to estimate the level of persisters it is necessary to perform a time-kill assay - biphasic killing curves should be observed. Moreover, the emergence of antibiotic-resistant mutants should be ruled out by re-exposing of the surviving cells to the same antibiotic. According to the definitions described in ref. [41], the term persisters can not be used as a synonym of antibiotic tolerant cells (Line 165).
Minor comments
1.The authors often use the term “the total number of bacteria/cells”. It suggests that the sum of culturable, nonculturable and dead cells is taken into account, whereas the authors refer this term to the CFU counts before antibiotic treatment or heat stress. Therefore, it would be better to use “the initial CFU level” or “the total number of culturable bacteria”…
2.In the Introduction section or at the beginning of Results, it is necessary to better explain the difference between persister cells, DFs and VBNC bacteria. The “dormancy continuum” is mentioned (line 87) in the text, but the relation between VBNC and DFs is not sufficiently discussed, although some discrepancies concerning VBNC are described (lines 658-668).
3.Line 200 – the title should be removed
4.Lines 228 – 231 “…after 24 hours of growth with the subsequent dying of the population”, “populations died…”, “rapid dying”. Since some bacteria were transformed to antibiotic tolerant cells, nonculturable cells or DFs, it would be better to wrote that populations did not die but lost their culturability.
5.Line 349 –“ARCs” = antibiotic tolerant?
6.Lines 352-354 Please clarify: “An increase in the number of P cells on the 28th day in S. typhimurium, apparently, can be explained by the transition of surviving P cells into the dormant state (DFs), which is characterized by stress resistance, including to antibiotics”. Does this mean that P cells are less stress resistant than DFs? It would be good to rearrange the text and add here that heating reactivates DFs (but why not P cells?)
7.Regarding mixed, binary biofilms – how the surviving cells were distinguished ( E. coli from K. rhizophila and S. typhimurium from S. aureus.)
Author Response
The authors thank the reviewer for all the comments.
Comments and Suggestions for Authors
In the presented manuscript Pankratov et al investigated the formation of antibiotic tolerant (P, persisters) cells and anabiotic dormant forms (DFs) in planktonic populations and biofilms exposed to antibiotics and heat stress. The authors found the during the growth of the various Gram-negative and Gram-positive strains the number of thermo- and antibiotic-tolerant (P) cells initially decreased, and then, on the 28th day of growth, unexpectedly increased.Electron microscopy revealed the presence of DFs in the late 28-day biofilms. The Authors concluded that cytoplasm vitrification and other changes in cell ultrastructure were responsible for the transformation of P cells into DFs. Since heat stress can restore DFs growth, enhanced levels of surviving cells were observed at the end of the experiments. I think that to confirm these conclusions, another scenario should be excluded - the selection of spontaneous mutations that enable growth of antibiotic- and heat-resistant mutants or, in general, GASP (growth advantage in stationary phase) mutants. For example, GASP mutants appear in 10-day-old E. coli cultures (see DOI: 10.1126/science.7681219).
Answer: Dear Rewiever #2! Thank you very much for your suggestion to exclude another possibility, the selection of spontaneous mutations that enable growth of antibiotic- and heat-resistant mutants. This possibility was excluded logically. The possibility for any specific mutant is 10-8-9 per dividing cell/generation. Meantime 106 cells were applied to each glass filter. Hence, content of possible mutants (heat-tolerant or ABR) in grown biofilm population is negligible. Regarding GASP mutants we can say, that we seeded glass filters by cells of early stationary phase, not 10-day-old. An addition has been made to the text to address these comments.
It is not clear how the number of persisters was estimated. According to ref. [41], to estimate the level of persisters it is necessary to perform a time-kill assay - biphasic killing curves should be observed. Moreover, the emergence of antibiotic-resistant mutants should be ruled out by re-exposing of the surviving cells to the same antibiotic. According to the definitions described in ref. [41], the term persisters can not be used as a synonym of antibiotic tolerant cells (Line 165).
Answer: Thanks for drawing our attention to “persisters” versus “tolerant cells” definitions. In some cases term persisters can be used as a synonym of antibiotic tolerant cells (“Persister cells are simply a subpopulation of tolerant bacteria” P. 442 of [41]). In our case ABT cells WERE persisters, as they fulfilled the definitions by [41]. We had bi-phasic death kinetics (bimodal killing curve) in all experiments. AB (ciprofloxacin) concentration used was higher than its 10 MICs. We clearly deal with triggered persistence (P1) induced by starvation in biofilm culture. We can exclude mutants for the same reasons as we gave above.
Minor comments
- The authors often use the term “the total number of bacteria/cells”. It suggests that the sum of culturable, nonculturable and dead cells is taken into account, whereas the authors refer this term to the CFU counts before antibiotic treatment or heat stress. Therefore, it would be better to use “the initial CFU level” or “the total number of culturable bacteria”…
Answer: We agree with this remark. Since we used only the cultural method of counting CFU, in order to correct this point, we made an addition in the Materials and Methods section ‘2.2. Production of planktonic populations’, where we clarify that the term "Total number of cells" refers to the total "number of cultured cells".
- In the Introduction section or at the beginning of Results, it is necessary to better explain the difference between persister cells, DFs and VBNC bacteria. The “dormancy continuum” is mentioned (line 87) in the text, but the relation between VBNC and DFs is not sufficiently discussed, although some discrepancies concerning VBNC are described (lines 658-668).
Answer: The appropriate revision has been made.
- Line 200 – the title should be removed
Answer: Wrongly duplicated title was removed.
- Lines 228 – 231 “…after 24 hours of growth with the subsequent dying of the population”, “populations died…”, “rapid dying”. Since some bacteria were transformed to antibiotic tolerant cells, nonculturable cells or DFs, it would be better to wrote that populations did not die but lost their culturability.
Answer: The appropriate revision has been made.
5.Line 349 –“ARCs” = antibiotic tolerant?
Answer: The appropriate revision has been made.
- Lines 352-354 Please clarify: “An increase in the number of P cells on the 28th day in S. typhimurium, apparently, can be explained by the transition of surviving P cells into the dormant state (DFs), which is characterized by stress resistance, including to antibiotics”. Does this mean that P cells are less stress resistant than DFs? It would be good to rearrange the text and add here that heating reactivates DFs (but why not P cells?)
Answer: The appropriate revision has been made.
- Regarding mixed, binary biofilms – how the surviving cells were distinguished ( E. coli from K. rhizophila and S. typhimurium from S. aureus.)
Answer: Since the colonies of these bacteria differ in color, shape, size, and consistency, we easily determined their numbers in mixed cultures. In section 2.3 we added information about the composition of binary biofilms. We added two suggestions to paragraph 2.4. (Determination of the number of viable cells in biofilms) that clarify the methodology of counting colonies of different species on agar plates.

Round 2
Reviewer 1 Report
The manuscript has been improved
Reviewer 2 Report
The authors addressed all my comments. Therefore, the revised version of the manuscript can be accepted for publication.